# Seroprevalence analysis of SARS-CoV-2 in pregnant women along the first pandemic outbreak and perinatal outcome

Cecilia Villalaín[1,2], Ignacio Herraiz[1,2]*, Joanna Luczkowiak[2], Alfredo Pérez-Rivilla[3], María Dolores Folgueira[2,3], Inmaculada Mejía[1], Emma Batllori[1], Eva Felipe[1], Beatriz Risco[1], Alberto Galindo[1,2]°, Rafael Delgado[2,3]°

1 Fetal Medicine Unit – Maternal and Child Health and Development Network (Red SAMID-RD12/0026/0016), Department of Obstetrics and Gynecology, Hospital Universitario 12 de Octubre, Universidad Complutense de Madrid, Madrid, Spain, 2 Instituto de Investigación Hospital 12 de Octubre (imas12), Madrid, Spain, 3 Department of Clinical Microbiology Hospital Universitario 12 de Octubre, Universidad Complutense de Madrid, Madrid, Spain

° These authors contributed equally to this work.
* ignacio.herraiz@salud.madrid.org

**Data Availability Statement:** All relevant data are within the manuscript and its Supporting information files.

## Abstract

### Objectives

To evaluate the progression of the seroprevalence of SARS-CoV-2 in the pregnant population of the south of Madrid during the first wave of the COVID-19 pandemic. Secondarily we aimed to evaluate maternal and perinatal outcomes.

### Study design

Retrospective cohort study conducted at Hospital Universitario 12 de Octubre during weeks 10 to 19 of 2020, coinciding with the Spanish lockdown. We tested 769 serum samples obtained from routine serological testing during the first and third trimesters of pregnancy for specific IgG anti SARS-CoV-2 RBD and S proteins. RT-PCR tests were performed in suspected cases according to clinical practice. We compared maternal and perinatal outcomes in those with delivered pregnancies (n = 578) according to the presence or absence of specific IgG antibodies. Those with positive IgG were subdivided by the presence or absence of Covid-19 related symptoms at any time and the results of RT-PCR testing if performed. Therefore, we had 4 study groups: G1 (IgG negative), G2 (IgG positive, asymptomatic, RT-PCR testing negative or not done), G3 (IgG positive, symptomatic, RT-PCR testing negative or not done), and G4 (IgG positive, symptomatic, RT-PCR positive).

### Results

Seropositivity increased from 0% to 21.4% (95% CI 11.8–31.0) during the study period, of which 27.9% had an asymptomatic course. Overall outcomes were favorable with a significant increased rate of preterm birth in G4 vs G1 (21.4% vs 6.7%) and cesarean/operative delivery (50% vs 26.9%). Asymptomatic and mild cases did not have differences regarding

**Funding:** This work was supported by grants to RD by the Instituto de Investigación Carlos III (grants FIS PI 1801007, by the European Union Commission Horizon 2020 Framework Programme: Project VIRUSCAN FETPROACT-2016: 731868. and by Fundación Caixa- Health Research (Project StopEbola). This work was supported by grant to MDF by the Instituto de Investigación Carlos III (FIS PI1800740). The funders had no role in study design, data collection and analysis, decision to publish, or preparation of the manuscript.

**Competing interests:** The authors have declared that no competing interests exist.

pregnancy course when compared to seronegative women. There were no documented cases of vertical or horizontal transmission.

## Conclusion

Seroprevalence in pregnant women in southern Madrid went up to 21.4% of which 27.9% had an asymptomatic course. Overall perinatal results were favorable, especially in those asymptomatic.

## Introduction

The novel human coronavirus SARS-CoV-2 has been responsible for a global pandemic during 2020 with over 27 million confirmed cases and over 891000 deaths in more than 180 countries [1]. Spain has been one of the most severely affected by the first wave of SARS-CoV-2 pandemic with an official report of 218000 cases confirmed by real-time PCR (RT-PCR) on nasopharyngeal swabs (NFS) during the first wave, by May 10th 2020 [2]. However, the national seroepidemiological study, conducted from April 27th to May 11th, suggested that the real number of infections was about 10 times higher [3]. This gap was justified by the lack of detection of asymptomatic cases or even the incomplete ascertainment of those with symptoms during the first wave of the pandemic. This study also showed that, within Spain, Madrid was particularly affected, reaching a seroprevalence of 11.5% [4]. The activity of the infection was not uniform even within cities, with wide differences in the infection rates described in their districts.

The SARS-CoV-2 infection in pregnancy has direct and indirect consequences [5, 6]. Focusing on the direct impact, current knowledge is mainly derived of cases diagnosed by RT-PCR in women with suggestive symptoms or in those asymptomatic that are hospitalized for non-COVID-19 related reasons [7]. This shows that first, maternal complications appear to be similar to non-pregnant women in the same age group, although pregnant women might be at increased risk of admission to an intensive care unit, second, there is an increased risk of preterm birth and cesarean section, and third, vertical transmission is considered rare [8, 9]. However, the full spectrum of SARS-CoV-2 infection remains unknown because mild or asymptomatic infections in non-hospitalized women are underdiagnosed. This could be overcome by studying the seroprevalence in the population of pregnant women, which can yield key data for health care providers in the event of a second wave or a future pandemic.

The main objective of our study is to report the seroprevalence of SARS-CoV-2 specific antibody in stored serum samples from the standard follow up of pregnancies during weeks 10–19 in 2020, coinciding with the onset, peak and decline of the first wave of the pandemic in Madrid. Secondarily we aim to evaluate maternal and perinatal outcomes stratified by the presence of antibodies against SARS-CoV-2, COVID-19 related symptoms and the result of RT-PCR on NFS.

## Materials and methods

This study was approved by the Institutional Review Board (Research Ethics Committee of the Hospital Universitario 12 de Octubre) and informed consent was obtained from all participants (N°:20/241).

## Study design and population

We present a retrospective cohort study conducted at Hospital Universitario 12 de Octubre, a large teaching hospital in the south of Madrid covering an estimated population of 400.000 inhabitants, being the reference center for 4000 deliveries/year.

Antenatal routine care includes two serological screenings for HIV, syphilis, rubella, hepatitis B, hepatitis C and toxoplasma congenital infection during pregnancy, one in the first trimester (between 8–12 weeks) or at the first visit in pregnancy, and a second one during the third trimester (between 32–35 weeks). These analyses are performed in our clinical microbiology laboratory with the exception of first trimester screening in low-risk pregnancies that is carried out at peripheral units. Our cohort was composed by pregnant women with these routine serological analysis between February 28th to May 10th (weeks 10–19 of 2020), from the beginning to the end of lockdown in Spain.

Serum samples are stored for at least a year at the biobank of the Microbiology Department for any additional testing that may be required in the follow-up of the mother, fetus or newborn. We performed an analysis of stored serum samples obtained during the study period for specific IgG anti SARS-CoV-2 RBD and S proteins according to the protocol described by Amanat et al [10]. SARS-CoV-2 seropositivity was defined as the presence of anti-SARS-CoV-2 IgG antibodies in serum (titer >1:100), and was considered as a marker of SARS-CoV-2 infection. All tested women had complete maternal and perinatal follow-up.

The patients were informed of this seroprevalence study through telephone communication, to minimize the risk of contagion inherent in a hospital visit in a pandemic situation. It was explained to them that the study was carried out in the context of the public health need to understand the seroprevalence in our environment and to verify the efficacy of a new test recommended by the World Health Organization (WHO) in the network of seroprevalence studies Solidarity II. Their verbal consent was obtained for participation in the study and to be informed of the results, all being reflected in the clinical history. In the cases where the result was positive, a hospital visit was also arranged to provide additional information on recommendations and follow-up, and their written informed consent for their participation in the study was also obtained.

## SARS-CoV-2 management

**Screening.** Since the report of the first case of COVID-19 in pregnant woman in Madrid on March 3rd, antenatal visits (both telephone and face-to-face consultations) include a systematic detailed questionnaire regarding most known SARS-CoV-2 infection symptoms, including fever, dry cough, sneezing, dyspnea, myalgia/asthenia, diarrhea, conjunctivitis, anosmia or ageusia. When any of the symptoms is present either during an antenatal visit or when consulting at the emergency room, women are tested by specific SARS-CoV-2 RT-PCR on NFS [7].

From March 31st to April 7th, it was decided to perform RT-PCR testing to all pregnant women with a scheduled cesarean section or labor induction, and since April 8th we also perform a universal screening with NFS on all women admitted to our maternity.

**RT-PCR confirmed case.** Management of pregnant women with a RT-PCR positive for SARS-CoV-2 includes a complete physical exam, blood work-up, chest X-ray if $SpO_2$ in pulse oximetry is below 96% or tachypnea, and fetal ultrasound. In absence of signs of pneumonia or in mild forms [11], prophylactic low weight heparin treatment for two weeks is initiated, and out-patient management is proposed with telephone follow-up every 48 hours until 14 days after diagnosis. If there are signs of clinical deterioration during follow-up the patient is asked to return to the emergency room for re-evaluation. If the patient's status does not allow

out-patient monitoring she is hospitalized and treatment with hydroxychloroquine and pro-phylactic heparin is initiated, according to our hospital's infectious disease protocol.

Delivery for SARS-CoV-2 reasons is only indicated when maternal deterioration requires considering mechanical ventilatory support. The preferable way for delivery is vaginal in absence of contraindications. Regardless of the SARS-CoV-2 status, skin-to-skin is not offered if delivery is via cesarean section given the lack of personnel to help the mothers in the process.

RT-PCR NFS testing is performed on all newborns of mothers with confirmed RT-PCR and in those with symptoms suggestive of SARS-CoV-2 infection.

**Serological confirmed case.**   Clinicians and patients are aware of the results of the anti-body study. Cases with a serological confirmation of SARS-CoV-2 infection, either with a his-tory of a positive result of RT-PCR on NFS or asymptomatic with no RT-PCR testing, are considered non-contagious, not recommending isolation and no special measures are taken for their management.

## Statistical analysis and data management

The main outcome was to study the seroprevalence of IgG specific antibodies in a cohort of pregnant women during the first wave of the pandemic. For our secondary analysis we strati-fied the population according to the results of the serological study (IgG positive vs. negative). Those with positive IgG were further subdivided depending on the presence or absence of COVID-19 related symptoms at any time before the serological study, and the results of the RT-PCR if performed. We analyzed 4 study groups: G1 (negative IgG), G2 (positive IgG, asymptomatic, negative or not done RT-PCR testing), G3 (positive IgG, symptomatic, negative or not done RT-PCR testing), and G4 (positive IgG, symptomatic, positive RT-PCR). For this analysis we excluded women who developed symptoms and had a subsequent positive RT-PCR after their serological testing. If the patient had mild symptoms but both RT-PCR and antibody testing were negative she was included in G1. G1 was considered the reference group for comparisons.

We recorded in all cases the following independent variables: (1) Main maternal basal char-acteristics and risk factors that have been associated to SARS-CoV-2 infection severity, includ-ing maternal age, body mass index, self-reported race, prior medical comorbidities and smoking; (2) Triage data: from March 3rd at every antenatal visit and at the emergency room women were asked for suggestive symptoms of SARS-CoV-2 infection. Any suggestive symp-toms recorded in the clinical history during the month of February were included as well. At triage of symptomatic women the following variables were recorded: temperature, fever (con-sidered as an axillary temperature $\geq 37.5°C$), peripheral oxygen saturation ($SpO_2$), lympho-cyte, leucocyte, platelet, transaminases and LDH levels and chest X-ray result (normal/abnormal); (3) Maternal outcomes: need for admission, length of admission, need for oxygen support, type of oxygen support and intensive care unit; (4) Perinatal outcomes in women with completed pregnancies (n = 579): first or second trimester miscarriage, stillbirth (fetal death >22 weeks), neonatal death (death of alive newborn <28 days) preterm birth <37 weeks, small for gestational age (estimated fetal weight <10th centile), intrauterine growth restriction [12], ultrasound morphological anomalies (evaluated at least at the first, second and third trimester), mode of delivery, reason for operative or cesarean delivery, neonatal acidosis defined as an arterial umbilical cord pH<7.1, Apgar at 5 min, neonatal intensive care unit admission, neonatal complications, evidence of vertical transmission, skin-to-skin and breast-feeding. All independent variables were analyzed between subjects.

Outcome data were recorded in a database created on the Research Electronic Data Capture (REDCap) tool [22] hosted at the "imas12" research institute. The Strengthening the Reporting

of Observational Studies in Epidemiology (STROBE) statement was followed for reporting the results. Continuous variables were expressed in mean (SD) or median (interquartile range) when non-normally distributed as assessed by Shapiro-Wilks´ test. Categorical variables were expressed in percentage (%). Univariate comparisons were performed between all groups and G1 using the t-test or Mann–Whitney U-test for continuous variables and the chi-square or Fisher's exact test for categorical variables. Two-sided P<0.05 was considered statistically significant. Data were carefully entered and analyzed after data cleansing, using statistical package STATA 14.2 (TX, USA: StataCorp LP).

The minimal anonymized data set necessary to replicate the study findings is included as a S1 Database.

## Results

During the study period, 769 women were tested for SARS-CoV-2 specific antibodies. There were 97, 51 and 621 serological tests performed on serum samples from the stored first, second and third trimester, respectively. A total of 86/769 (11.2%) samples resulted positive for SARS-CoV-2 RBD-specific IgG. The first positive sample was obtained at week 11 and the proportion of positive samples increased over time to reach 23.3% (CI 95% 13.6–33.0) and 21.4% (95% CI 11.8–31.0) by weeks 18 and 19 respectively, as shown in Fig 1.

There were 92/769 (11.9%) women who referred COVID-19 related symptoms at some time from February, of whom 36/92 (39.1%) were tested by RT-PCR and 31/36 (86.1%) were positive. All cases confirmed by RT-PCR had positive antibodies except for 10 cases in which the serological test had been performed prior to the onset of symptoms. These 10 cases were excluded for further analysis.

G1 was composed by 673/759 (88.7%) seronegative women. Of note, 23/673 (3.1%) referred at least one COVID-19 related symptom. G2 comprised 24/759 (3.1%) asymptomatic seropositive women. G3 was constituted by 40/759 (5.3%) symptomatic seropositive women in which confirmative RT-PCR was not performed since they never attended the emergency department for their NFS RT-PCR testing (n = 36), or the RT-PCR was negative (n = 6). Finally, G4 consisted of 22/759 (2.9%) symptomatic seropositive women with RT-PCR confirmation. The distribution of the different study groups is described in Fig 2.

The main basal characteristics of the study population according to the serological status are depicted in Table 1.

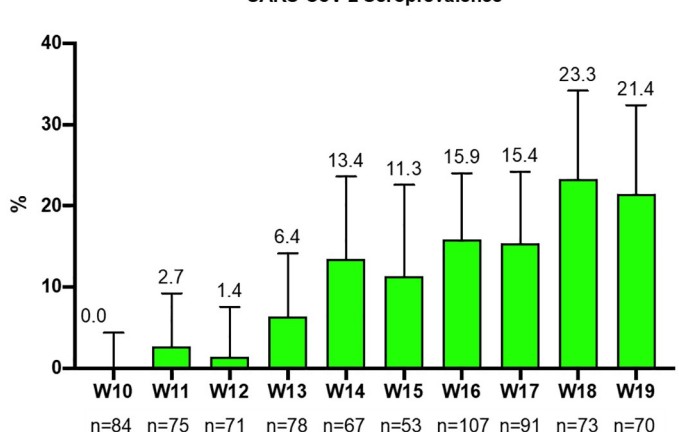

**Fig 1. SARS-CoV-2 seroprevalence during weeks 10–19 of 2020.**

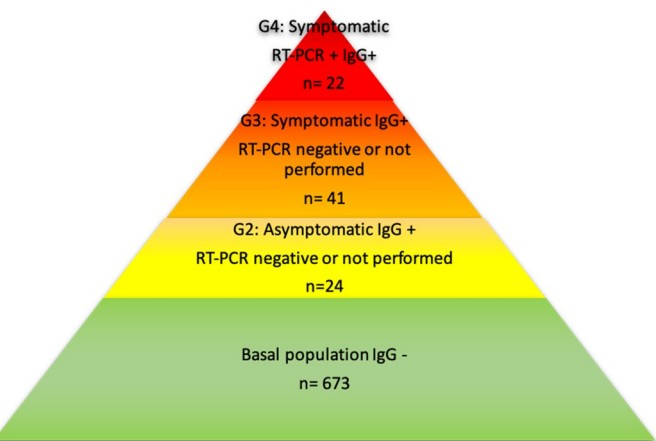

**Fig 2. Population distribution according to their microbiological status and symptoms.** *10 cases with positive
PCR and symptoms after serological evaluation were excluded.

As shown, there were not statistically significant differences among them except for a lower
height and a higher BMI in G4 when compared to G1. As expected by the definition of the
groups there were also differences regarding the presence of COVID-19 related symptoms
which were higher in G3 and G4. The most frequent symptoms in G3 and G4 were dry cough
(48.8% and 68.2%, respectively) followed by myalgia (48.8% and 68.2%) and fever (45.0% and
54.6%).

Among the 86 seropositive women, 24 (27.9%) were asymptomatic, while 13 (15.1%)
required admission, all of them for respiratory symptoms in the context of pneumonia with a
pathological chest X-ray. Median hospital stay was of 5 days (range 1–11) and none of them
required intensive care unit admission.

There was a total of 578 women with complete perinatal outcomes, depicted in Table 2.
Median gestational age at delivery was 39.7 weeks (range 27.0–41.7). There were two cases of
perinatal mortality in the seronegative group. The overall results showed non-significant dif-
ferences between G2 or G3 and the basal group (G1). However, we found an increased risk of
preterm birth among G4 when compared to G1 (21.3 *vs* 6.7, p = 0.03). The three cases of pre-
term birth in G4 were not directly related to a COVID-19 complication: one case was a mono-
chorionic monoamniotic pregnancy in which a caesarean section was scheduled at 32 weeks
and the other two were cases of preterm rupture of membranes in which labour induction was
scheduled at 34 weeks according to local guidelines. We also observed a higher rate of opera-
tive delivery (either cesarean or operative vaginal) in G4 vs G1 (50% *vs* 26.9%, p = 0.01) with
non-significant differences in the operative delivery rate for fetal distress (42.9% *vs* 24.8%,
p = 0.28). There were no reported cases of suspected vertical nor horizontal transmission.

## Discussion

### Main findings of the study

Our study highlights the impact of the first wave of the SARS-CoV-2 pandemic on the preg-
nant women population in Madrid, during weeks 10–19 of 2020. The seroprevalence analysis
performed in archived serum samples reflects the epidemic trend reaching over 21.3% sero-
positivity by week 19. The overall perinatal results of seropositive women were favorable and
up to 27.9% of infected women had an asymptomatic course. We found no cases of vertical
transmission in our cohort.

**Table 1. Maternal baseline characteristics.**

| | G1 | G2 | G3 | G4 | p |
|---|---|---|---|---|---|
| | n = 673 | n = 24 | n = 40 | n = 22 | |
| Maternal age in years | 32 (11) | 31.5 (8) | 30 (10) | 29.5 (11) | 0.88 |
| Height in cm | 162 (10) | 160 (14) | 163 (10) | 159 (10) | 0.26 |
| Weight in kg | 62 (16) | 64 (11) | 69 (14)* | 70 (12)* | 0.11 |
| Body mass index in kg/m$^2$ | 23 (5) | 25 (5) | 24.5 (7) | 26 (6.5) | <0.01 |
| Nulliparous | 308 (45.8) | 10 (41.7) | 15 (37.5) | 10 (45.5) | 0.76 |
| Race | | | | | |
| •Non-hispanic white | 353 (52.5) | 11 (45.8) | 19 (46.3) | 10 (45.5) | |
| •Black | 13 (2.0) | 0 (0) | 3 (7.3) | 0 (0) | |
| •Hispanic | 233 (34.6) | 12 (50.0) | 16 (40.0) | 12 (54.6) | 0.30 |
| •North African | 41 (6.1) | 1 (4.2) | 0 (0) | 0 (0) | |
| •Asian | 31 (4.6) | 0 (0) | 2 (4.9) | 0 (0) | |
| •Other / unknown | 2 (0.3) | 0 (0) | 0 (0) | 0 (0) | |
| Active smoking | 39 (5.8) | 2 (8.3) | 3 (7.7) | 0 (0) | 0.32 |
| Chronic medical pathology | | | | | |
| •None | 554 (82.3) | 17 (70.8) | 34 (85.0) | 17 (77.3) | 0.44 |
| •Cardiovascular / cerebrovascular | 14 (2.1) | 1 (4.2) | 0 (0) | 2 (9.1) | 0.11 |
| •Respiratory system | 15 (2.2) | 1 (4.2) | 2 (4.9) | 2 (9.1) | 0.17 |
| •Digestive system | 10 (1.5) | 0 (0) | 1 (2.4) | 0 (0) | 0.81 |
| •Endocrine system | 31 (4.6) | 2 (8.3) | 4 (9.8) | 0 (0) | 0.27 |
| •Malignant tumor | 1 (0.2) | 0 (0) | 0 (0) | 0 (0) | 0.99 |
| •Nervous system | 12 (1.8) | 0 (0) | 0 (0) | 0 (0) | 0.67 |
| •Others | 79 (11.7) | 4 (16.7) | 4 (9.8) | 2 (9.1) | 0.83 |
| Symptoms | | | | | |
| •Asymptomatic | 652 (96.9) | 24 (100) | 0* | 0* | <0.01 |
| •Fever | 12 (0.2) | 0 (0) | 18 (45.0)* | 12 (54.6)* | <0.01 |
| •Dry cough | 13 (0.2) | 0 (0) | 20 (48.8)* | 15 (68.2)* | <0.01 |
| •Sneezing | 5 (0.01) | 0 (0) | 6 (14.6) | 14 (63.6)* | <0.01 |
| •Myalgia/asthenia | 8 (0.01) | 0 (0) | 19 (48.7)* | 15 (68.2)* | <0.01 |
| •Anosmia/ageusia | 1 (0.0) | 0 (0) | 18 (30)* | 9 (40.9)* | <0.01 |
| •Conjunctivitis | 1 (0.0) | 0 (0) | 2 (5.4)* | 1 (4.6)* | <0.01 |
| •Diarrhea | 1 (0.0) | 0 (0) | 7 (17.5)* | 5 (22.7)* | <0.01 |

Data presented as median (interquartile range) and n (%)

*Statistically significant differences when compared to G1

†Bonferroni correction was applied

G1: IgG negative; G2: IgG positive, asymptomatic, RT-PCR testing negative or not done; G3: IgG positive, symptomatic, RT-PCR testing negative or not done; G4: IgG positive, symptomatic, RT-PCR positive.

## Interpretation of results and comparison with existing literature

The impact of SARS-CoV-2 in pregnancy has been mainly assessed from the perspective of the cases detected by RT-PCR. As far as we know, seroprevalence of COVID-19 in pregnant woman has been only reported once, in a multicenter study carried out in Barcelona in the last phase of the first wave (April 14 to May 5, 2020) and in which 14.0% were positive for anti-SARS-CoV-2 [13]. The seropositive rate for Madrid and Barcelona obtained through random sample participants by national health authorities in Spain has been of 11.5% and 6.8%, respectively [4]. Therefore, the percentages of infection obtained in pregnant women are higher in

**Table 2. Perinatal outcomes stratified by serological results and symptoms.**

| | G1 | G2 | G3 | G4 | p |
|---|---|---|---|---|---|
| | n = 523 | n = 16 | n = 24 | n = 14 | |
| **First trimester miscarriage** | 6 (1.2) | 0 (0) | 0 (0) | 0 (0) | 0.89 |
| **Second trimester miscarriage**[a] | 2 (0.4) | 0 (0) | 0 (0) | 0 (0) | 0.97 |
| **Stillbirth**[b] | 2 (0.4) | 0 (0) | 0 (0) | 0 (0) | 0.98 |
| **Pregnancy medical complications**[b] | | | | | |
| •None | 457 (89.0) | 15 (93.8) | 23 (95.8) | 10 (71.4) | 0.12 |
| •Hypertensive disorder | 11 (2.1) | 1 (6.3) | 0 (0) | 1 (7.1) | 0.16 |
| •Gestational diabetes | 31 (6.1) | 0 (0) | 1 (4.2) | 1 (7.1) | 0.74 |
| •Other | 22 (4.3) | 1 (5.9) | 0 (0) | 2 (14.3) | 0.21 |
| **Small for gestational age**[b] | 14 (2.7) | 0 (0) | 1 (4.2) | 1 (7.1) | 0.66 |
| **Intrauterine growth restriction**[b] | 17 (3.3) | 0 (0) | 0 (0) | 0 (0) | 0.59 |
| **Fetal anomalies at standard ultrasound**[a] | 20 (3.9) | 0 (0) | 0 (0) | 1 (7.1) | 0.54 |
| **Gestational age at delivery in weeks**[c] | 39.9 (1.9) | 39.6 (2.1) | 39.5 (2.7) | 38.5 (3.3) | 0.95 |
| **Preterm birth**[c] | 34 (6.7) | 1 (5.9) | 0 (0) | 3 (21.4)* | 0.16 |
| **Mode of delivery**[c] | | | | | |
| •Eutocic | 376 (73.3) | 10 (62.5) | 18 (75.0) | 7 (50.0) | |
| •Operative | 28 (5.5) | 2 (11.8) | 1 (4.2) | 3 (21.4) | 0.69 |
| •Cesarean section | 109 (21.2) | 4 (23.5) | 5 (20.8) | 4 (28.6) | |
| **Cesarean or operative delivery for fetal distress**[c] | 34 (24.8) | 1 (16.7) | 3 (50.0) | 3 (42.9) | 0.29 |
| **Birthweight in g**[c] | 3250 (620) | 3200 (560) | 3430 (650) | 3270 (810) | 0.36 |
| **Apgar score at 5 min <7**[c] | 4 (0.9) | 0 (0) | 1 (2.7) | 0 (0) | 0.68 |
| **Arterial pH <7.1**[c] | 14 (2.7) | 0 (0) | 0 (0) | 1 (7.1) | 0.63 |
| **Neonatal intensive care unit admission**[c] | 28 (5.5) | 0 (0) | 0 (0) | 1 (7.1) | 0.48 |
| **Neonatal SARS-CoV-2 infection**[c] | 0 (0) | 0 (0) | 0 (0) | 0 (0) | NA |
| **Skin to skin**[c] | 402 (79.3) | 12 (75.0) | 22 (91.7) | 10 (71.4) | 0.40 |
| **Breastfeeding**[c] | 494 (97.2) | 16 (100) | 24 (100) | 13 (92.9) | 0.53 |

NA, not applicable

Data presented as median (interquartile range) and n (%)

*Statistically significant differences when compared to G1

[a] Calculated over completed pregnancies excluding 1st trimester miscarriages (n = 567)

[b] Calculated over completed pregnancies excluding 1st and 2nd trimester miscarriages (n = 565)

[c] Calculated over completed pregnancies with live newborns (n = 563)

G1: IgG negative; G2: IgG positive, asymptomatic, RT-PCR testing negative or not done; G3: IgG positive, symptomatic, RT-PCR testing negative or not done; G4: IgG positive, symptomatic, RT-PCR positive.

both studies. Nevertheless it is possible that the distribution of cases even within different city areas could be significantly different as has been documented for New York City [14] and the finding of a higher infection rate of pregnant women could be just a reflection of the infection rate in the area of the city that has been studied. Additionally, our study provides information about the evolution of seropositivity, showing that there is a delay of 2–3 weeks in its increment with respect to that observed in cases diagnosed by RT-PCR, until it becomes stable from week 18 [15]. This is in accordance with the expected time to develop IgG antibodies after detection of SARS-CoV-2 by RT-PCR.

Interestingly 28% of infected women were asymptomatic in our study, and up to 60% in the study from Barcelona. This could be explained, at least partially, by our lower proportion of samples from the first trimester (43% *vs* 13%), since they found a significantly lower prevalence

of symptomatic infection and hospital admissions in women in their first trimester than those in their third trimester. Pregnant women are usually healthy patients that frequently visit health care centers and therefore should be considered as a possible source of infection transmission to more vulnerable cohorts that also attend these areas. This asymptomatic aspect of SARS-CoV-19 infection in pregnant women has been described in other series in US [16, 17] and Europe [18, 19] where universal screening upon hospital admission has been proposed for better resource allocation and protection of the woman's contacts including other patients or health-care professionals.

Our data show good overall maternal and perinatal outcomes in seropositive women, regardless of their symptoms. We stratified seropositive women in three groups according to their symptoms and RT-PCR confirmation. Pregnant women with symptomatic COVID-19 had higher BMI than non-infected women as has been described in non-obstetric population [20], and reflecting the importance of prevention in the specific subgroup of the obese gravida. It is of interest the increased preterm birth rate when comparing the most symptomatic cases with RT-PCR confirmation (G4) with the reference population (G1) as well a higher rate of operative delivery. Our small sample size is limited to draw conclusions about the implication of SARS-CoV-2 infection in these findings, although these two outcomes had already been described in other cohorts of pregnant women with RT-PCR confirmed SARS-CoV-2 infection [7]. Almost 2/3 of seropositive women had symptoms suggestive of SARS-CoV-2 infection but did not undergo RT-PCR testing, mainly for two reasons. Firstly, these were milder cases where the patients did not want to attend the emergency room for testing as they consider this as a non-safe place at that time. Secondly, most of the times when they referred these symptoms during their antenatal visit, they had already resolved and therefore it was considered unnecessary to test them given the poorer performance of RT-PCR if performed later on the disease [21]. Their maternal results were excellent since none of them required hospital admission for support and we found no significant differences regarding perinatal outcomes. Although we only have a small sample of asymptomatic seropositive women, their maternal and perinatal results overlap with those of uninfected pregnant women.

We had no documented cases of vertical nor horizontal transmission, accordingly to recently published studies [8, 22], which reinforces our policy of promoting skin-to-skin and breastfeeding, given their known benefits for both mother and child.

## Strengths and limitations

We acknowledge some limitations in our study. The results are drawn from a cohort of women from the south of Madrid, and cannot be extrapolated to other communities. Moreover, we could not analyze those pregnancies delivered before reaching the third trimester serology during the study period. RT-PCR on NFS was not systematically performed due to the abovementioned patients' concerns as well as the progressive implementation of this test during the pandemic [17]. Finally, this study was not designed to evaluate pregnancy outcomes and it may lack power to evaluate some events, especially in the most underrepresented group which is seropositive asymptomatic women.

However, there are also several strengths. This seroprevalence study provides additional information in an issue that has been scarcely explored. Moreover, it covers the whole first wave of the pandemic showing nicely the evolution of the seroprevalence in our population. Finally, it has allowed us to evaluate the rate of asymptomatic pregnant women affected by SARS-CoV-2 and to obtain a first glimpse into its consequences in pregnancy.

### Clinical perspective

Our data can help to better understand the evolution of the pandemic both in Madrid and worldwide, with the finding that up to 28% of infected patients had no symptoms at all. Assessing the cumulative prevalence of SARS-CoV-2 infection can help trace contacts, identify hosts with viral reservoir, estimate viral propagation in certain communities, know the rate of asymptomatic infections and obtain better estimates on both morbidity and mortality.

These results can also help obstetricians when counseling pregnant women. Although data are still limited, we can reassure them of the good maternal and perinatal results, especially in those with an asymptomatic course. We also found no evidence of impaired fetal growth, fetal anomalies related to the infection nor any signs of vertical nor horizontal transmission, which have been some of the major concerns in pregnancy care.

From an epidemiological-public health point of view, future research should focus in the follow-up of antibody levels and their efficacy in any upcoming waves. This information will be key for health-care system planning in any future events.

From a perinatal view, further investigation should focus first, in the impact of the SARS-CoV-2 infection in pregnancy at any trimester and secondly, in the follow-up of infected mothers and their newborns given the lack of knowledge on the infant's development or long-term outcomes for the mothers.

## Conclusions

A high prevalence of COVID-19 has been detected in pregnant women in Madrid, Spain, during the outbreak of the first epidemic wave. The asymptomatic profile of 27.9% of cases warrants specific protocols for diagnosis and could have important clinical and epidemiological implications. Maternal and perinatal outcomes seem favorable, especially in those asymptomatic or with milder presentation. Nevertheless SARS-CoV-2 infection warrants specific follow up for both mothers and newborns.

## Supporting information

**S1 Database.**
(CSV)

## Acknowledgments

We would like to express our gratitude to everybody in our Obstetrics and Gynecology Department for their great effort during the pandemic.

## Author Contributions

**Conceptualization:** Cecilia Villalaín, Ignacio Herraiz, Alfredo Pérez-Rivilla, María Dolores Folgueira, Inmaculada Mejía, Rafael Delgado.

**Data curation:** Cecilia Villalaín, Joanna Luczkowiak, Alfredo Pérez-Rivilla, María Dolores Folgueira, Inmaculada Mejía, Emma Batllori, Eva Felipe, Beatriz Risco, Rafael Delgado.

**Formal analysis:** Cecilia Villalaín, Ignacio Herraiz, Alfredo Pérez-Rivilla, Inmaculada Mejía, Rafael Delgado.

**Investigation:** Cecilia Villalaín, Joanna Luczkowiak, Inmaculada Mejía, Emma Batllori, Beatriz Risco, Alberto Galindo.

**Methodology:** Cecilia Villalaín, Ignacio Herraiz.

**Project administration:** Cecilia Villalaín, Ignacio Herraiz, Joanna Luczkowiak, Emma Batl-lori, Alberto Galindo.

**Supervision:** Cecilia Villalaín, Ignacio Herraiz, Alberto Galindo.

**Validation:** Cecilia Villalaín, Ignacio Herraiz, Joanna Luczkowiak, Eva Felipe, Beatriz Risco, Alberto Galindo.

**Writing – original draft:** Cecilia Villalaín.

**Writing – review & editing:** Cecilia Villalaín, Ignacio Herraiz, Alfredo Pérez-Rivilla, María Dolores Folgueira, Alberto Galindo, Rafael Delgado.

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
