## [Decision Letter · Decision Letter 0]

27 Oct 2020

PONE-D-20-28445

Seroprevalence analysis of SARS-CoV-2 in pregnant women along the pandemic outbreak

PLOS ONE

Dear Dr. Herraiz,

Thank you for submitting your manuscript to PLOS ONE. After careful consideration, we feel that it has merit but does not fully meet PLOS ONE’s publication criteria as it currently stands. Therefore, we invite you to submit a revised version of the manuscript that addresses the points raised during the review process.

We look forward to receiving your revised manuscript.

Kind regards,

Rogelio Cruz-Martinez, Ph.D.

Academic Editor

PLOS ONE

Journal Requirements:

a) Did participants provide their written or verbal informed consent to participate in this study?

3. For more information on PLOS ONE's expectations for statistical reporting, please see https://journals.plos.org/plosone/s/submission-guidelines.#loc-statistical-reporting. Please update your Methods and Results sections accordingly.

4. Thank you for stating the following in the Funding Section of your manuscript:

"This work was supported by grants to RD by the Instituto de Investigación Carlos III (grants FIS

 PI 1801007, by the European Union Commission Horizon 2020 Framework Programme: Project

VIRUSCAN FETPROACT-2016: 731868. and by Fundación Caixa-Health Research (Project

StopEbola). This work was supported by grant to MDF by the Instituto de Investigación Carlos III

(FIS PI1800740). The funders had no role in study design, data collection and analysis, decision

to publish, or preparation of the manuscript."

Reviewers' comments:

Reviewer's Responses to Questions

**Comments to the Author**

1. Is the manuscript technically sound, and do the data support the conclusions?

Reviewer #1: Yes

Reviewer #2: Yes

Reviewer #3: Yes

2. Has the statistical analysis been performed appropriately and rigorously? 

Reviewer #1: Yes

Reviewer #2: Yes

Reviewer #3: I Don't Know

3. Have the authors made all data underlying the findings in their manuscript fully available?

Reviewer #1: Yes

Reviewer #2: Yes

Reviewer #3: Yes

4. Is the manuscript presented in an intelligible fashion and written in standard English?

Reviewer #1: Yes

Reviewer #2: Yes

Reviewer #3: Yes

5. Review Comments to the Author

Reviewer #1: Overall, the results of this study are very timely and relevant. There are in my opinion a few of issues that need to be clarified/addressed.

Is the highest number of preterm and cesarean births due to SARS-CoV2 positive RT-PCR? maternal respiratory impairment, obstetric or fetal cause? Can you discuss this point?

Reviewer #2: The manuscript by Ignacio Herraiz et al. addressed the issue on Seroprevalence analysis of SARS-CoV-2 in pregnant women along the pandemic outbreak

The article is interesting: it is well written, the methodology used is consistent and results are important. They provide information in the the seroprevalence of SARS-CoV-2 in the pregnant population of the south of Madrid during the outbreak and in the evaluation of maternal

and perinatal outcomes.

There is currently uncertainty regarding the impact of the infection caused by the severe acute respiratory distress syndrome coronavirus 2 (SARS‐CoV‐2) in pregnancy .

I consider that the information provided by the authors is useful and should be known by the scientific community

TITLE:

The title is clear and appropriate. "Seroprevalence analysis of SARS-CoV-2 in pregnant women along the 1 pandemic outbreak" However, it can be clarified more in the following way "Seroprevalence analysis of SARS-CoV-2 in pregnant women along the 1 pandemic outbreak and perinatal outcome."

The abstract appropriately summarizes the manuscript without omitting important results.

INTRODUCTION

The topic is of clinical importance. Most of the previous work is adequately discussed and referenced. The rationale provided by the authors is sufficient and the purpose of the study is clearly defined.

However line 54 " Regarding the impact of SARS-CoV-2 infection in pregnancy " It is important to clarify that the impact can be direct or indirect (psychological stress and problems with public health programs)

please add these two references

Ref - Miguel Parra-Saavedra et al, Attitudes and collateral psychological effects of COVID‐19 in pregnant women in Colombia ,International Journal of Gynecology and Obstetrics 2020

Ref - Roberton T, Carter ED, Chou VB, et al. Early estimates of the indirect

effects of the coronavirus pandemic on maternal and child

mortality in low‐ and middle‐income countries. Lancet Glob Health.

2020;8:e901–e908.

METHODS

The methodology is appropriate and sufficient for the stated purpose.

Reviewer #3: Dr Herraiz et al made a good study of SARS-COV-2 seroprevalence, their results are in line with previous reports about the subject.

The only comment about this manuscript is: how would you explain the relationship between higher BMI and seropositivity and symptomatology (groups G1 vs G4)? This is described also in non-obstetric population, and highlighting this finding would add interesting information about SARS-COV-2 and pregnancy.

6. PLOS authors have the option to publish the peer review history of their article (what does this mean?). If published, this will include your full peer review and any attached files.

Reviewer #1: No

Reviewer #2: No

Reviewer #3: No

---

## [Author Response · Author response to Decision Letter 0]

8 Nov 2020

We have adjusted our manuscript to PLOS ONE´s style requirements. We have followed the instructions for file naming.

a) Did participants provide their written or verbal informed consent to participate in this study?

We have now amended the ethics statement to address these concerns:

- We have included the following statement a at the beginning of the Methods section as requested by the editors:

“This study was approved by the Institutional Review Board (Research Ethics Committee of the Hospital Universitario 12 de Octubre) and informed consent was obtained from all participants (Nº:20/241).”

- We have included the following additional information in the Methods section, line 91: 

“The patients were informed of this seroprevalence study through telephone communication, to minimize the risk of contagion inherent in a hospital visit in a pandemic situation. It was explained to them that the study was carried out in the context of the public health need to understand the seroprevalence in our environment and to verify the efficacy of a new test recommended by the World Health Organization (WHO) in the network of seroprevalence studies Solidarity II. Their verbal consent was obtained for participation in the study and to be informed of the results, all being reflected in the clinical history. In the cases where the result was positive, a hospital visit was also arranged to provide additional information on recommendations and follow-up, and their written informed consent for their participation in the study was also obtained.”

3. For more information on PLOS ONE's expectations for statistical reporting, please see https://journals.plos.org/plosone/s/submission-guidelines.#loc-statistical-reporting. Please update your Methods and Results sections accordingly.

 We have now updated our Methods and Results sections according to the PLOS ONE´s requirements. 

4. Thank you for stating the following in the Funding Section of your manuscript:

"This work was supported by grants to RD by the Instituto de Investigación Carlos III (grants FIS PI 1801007, by the European Union Commission Horizon 2020 Framework Programme: Project VIRUSCAN FETPROACT-2016: 731868. and by Fundación Caixa-Health Research (Project StopEbola). This work was supported by grant to MDF by the Instituto de Investigación Carlos III (FIS PI1800740). The funders had no role in study design, data collection and analysis, decision to publish, or preparation of the manuscript."

Yes, thank you for your support. We have now removed the funding-related text from the manuscript and included such statement in the cover letter.

Thank you again for your support. We have now uploaded the minimal anonymized data set necessary to replicate our study findings as supporting information.

Reviewer #1: Overall, the results of this study are very timely and relevant. There are in my opinion a few of issues that need to be clarified/addressed.

Is the highest number of preterm and cesarean births due to SARS-CoV2 positive RT-PCR? maternal respiratory impairment, obstetric or fetal cause? Can you discuss this point?

Yes, indeed. As stated in the discussion, several studies on pregnant women suffering from COVID-19 with positive RT-PCR have described an excess in the number of caesarean sections and preterm births. Unfortunately, our limited sample size of pregnant women with symptoms and positive RT-PCR (G4, n=14) prevent us to provide further evidence in this regard. Although we have certainly observed a higher rate of prematurity in G4, we had only 3 cases. We have examined our three cases of preterm birth in G4: one case was a monochorionic monoamniotic pregnancy in which a caesarean section was scheduled at 32 weeks and the other two were cases of preterm rupture of membranes in which labour induction was scheduled at 34 weeks according to local guidelines. In the case of caesarean sections, our results were not statistically significant, probably due to the small sample size.

We have now included this detailed information in the results section (line 225).

Therefore, we are unable to draw firm conclusions, as we have now discussed. Accordingly, we have replaced the following sentence in the discussion section (line 281):

“These two outcomes had already been described in other cohorts of pregnant women with RT-PCR confirmed SARS-CoV-2 infection.”

By:

“Our small sample size is limited to draw conclusions about the implication of SARS-CoV-2 infection in these findings, although these two outcomes had already been described in other cohorts of pregnant women with RT-PCR confirmed SARS-CoV-2 infection [7].”

Reviewer #2: The manuscript by Ignacio Herraiz et al. addressed the issue on Seroprevalence analysis of SARS-CoV-2 in pregnant women along the pandemic outbreak

The article is interesting: it is well written, the methodology used is consistent and results are important. They provide information in the the seroprevalence of SARS-CoV-2 in the pregnant population of the south of Madrid during the outbreak and in the evaluation of maternal and perinatal outcomes.

There is currently uncertainty regarding the impact of the infection caused by the severe acute respiratory distress syndrome coronavirus 2 (SARS‐CoV‐2) in pregnancy.

I consider that the information provided by the authors is useful and should be known by the scientific community.

TITLE: The title is clear and appropriate. "Seroprevalence analysis of SARS-CoV-2 in pregnant women along the 1 pandemic outbreak" However, it can be clarified more in the following way "Seroprevalence analysis of SARS-CoV-2 in pregnant women along the 1 pandemic outbreak and perinatal outcome."

Thank you for your suggestion. We have changed the title accordingly.

The abstract appropriately summarizes the manuscript without omitting important results.

INTRODUCTION: The topic is of clinical importance. Most of the previous work is adequately discussed and referenced. The rationale provided by the authors is sufficient and the purpose of the study is clearly defined.

However line 54 " Regarding the impact of SARS-CoV-2 infection in pregnancy " It is important to clarify that the impact can be direct or indirect (psychological stress and problems with public health programs) please add these two references

Ref - Miguel Parra-Saavedra et al, Attitudes and collateral psychological effects of COVID‐19 in pregnant women in Colombia ,International Journal of Gynecology and Obstetrics 2020

Ref - Roberton T, Carter ED, Chou VB, et al. Early estimates of the indirect

effects of the coronavirus pandemic on maternal and child

mortality in low‐ and middle‐income countries. Lancet Glob Health.2020; 8: e901–e908.

Yes, indeed. We thank the reviewer for this important remark. We have incorporated the two references reflecting the indirect impact of COVID-19 in pregnant women. We have also replaced the sentence (Introduction, line 54):

“Regarding the impact of SARS-CoV-2 infection in pregnancy, the current knowledge is…”

By:

“The SARS-CoV-2 infection in pregnancy has direct and indirect consequences (two new references inserted here). Focusing on the direct impact, current knowledge is…”

METHODS. The methodology is appropriate and sufficient for the stated purpose.

Reviewer #3: Dr Herraiz et al made a good study of SARS-COV-2 seroprevalence, their results are in line with previous reports about the subject.

The only comment about this manuscript is: how would you explain the relationship between higher BMI and seropositivity and symptomatology (groups G1 vs G4)? This is described also in non-obstetric population, and highlighting this finding would add interesting information about SARS-COV-2 and pregnancy.

Yes, thank you for arising this interesting point. We have highlighted this finding in the discussion section adding the following sentence (line 282):

“Pregnant women with symptomatic COVID-19 had higher BMI than non-infected women as has been described in non-obstetric population (new reference: Azzolino D, Cesari M. Obesity and COVID-19. Front Endocrinol (Lausanne). 2020;11:581356.), and reflecting the importance of prevention in the specific subgroup of the obese gravida”

---

## [Editor Report · Decision Letter 1]

16 Nov 2020

Seroprevalence analysis of SARS-CoV-2 in pregnant women along the first pandemic outbreak and perinatal outcome

PONE-D-20-28445R1

Dear Dr. Herraiz,

We’re pleased to inform you that your manuscript has been judged scientifically suitable for publication and will be formally accepted for publication once it meets all outstanding technical requirements.

Kind regards,

Rogelio Cruz-Martinez, Ph.D.

Academic Editor

PLOS ONE

Additional Editor Comments (optional):

The authors should be congratulated for the effort to publish this very-well designed study during this Sars-Cov2 pandemia

---

## [Editor Report · Acceptance letter]

18 Nov 2020

PONE-D-20-28445R1 

Seroprevalence analysis of SARS-CoV-2 in pregnant women along the first pandemic outbreak and perinatal outcome 

Dear Dr. Herraiz:

I'm pleased to inform you that your manuscript has been deemed suitable for publication in PLOS ONE. Congratulations! Your manuscript is now with our production department. 

Kind regards, 

on behalf of

Dr Rogelio Cruz-Martinez 

Academic Editor

PLOS ONE